# Marijuana Use May Be Associated with Reduced Prevalence of Prostate Cancer: A National Survey on Drug Use and Health Study from United States of America

**DOI:** 10.3390/biomedicines12051008

**Published:** 2024-05-03

**Authors:** Turab Mohammed, James Yu, Yong Qiao, Youngchul Kim, Eric Mortensen, Helen Swede, Zhao Wu, Jingsong Zhang

**Affiliations:** 1Department of Hematology-Oncology, H. Lee Moffitt Cancer Center and Research Institute, Tampa, FL 33612, USA; turab.mohammed@moffitt.org (T.M.); james.yu@moffitt.org (J.Y.); 2Department of Statistics, University of Connecticut, Storrs, CT 06269, USA; yong.qiao@uconn.edu; 3Department of Biostatistics and Bioinformatics, H. Lee Moffitt Cancer Center and Research Institute, Tampa, FL 33612, USA; youngchul.kim@moffitt.org; 4Department of Medicine, University of Connecticut School of Medicine, Farmington, CT 06032, USA; mortensen@uchc.edu; 5Department of Public Health Sciences, University of Connecticut School of Medicine, Farmington, CT 06032, USA; swede@uchc.edu; 6Department of Psychiatry, University of Connecticut Health Center, Farmington, CT 06030, USA; zwu@uchc.edu; 7Department of Genito-Urology Oncology, H. Lee Moffitt Cancer Center and Research Institute, Tampa, FL 33612, USA

**Keywords:** prostate cancer, cannabinoids, marijuana use, NSDUH, medical marijuana

## Abstract

Preclinical evidence indicates the potential anti-tumor capabilities of cannabinoids in prostate cancer (PC). We undertook a cross-sectional study using National Survey on Drug Use and Health data from 2002 to 2020, involving 2503 participants in the USA. The independent variable was marijuana use status (current, former, never), while the dependent variable was self-reported PC (yes, no). Eleven other demographic variables were assessed as covariates. PC prevalence was lower among current marijuana users (46/145, 31.7%) and former users (323/1021, 31.6%) compared to non-users (534/1337, 39.9%, *p* < 0.001). PC prevalence was lower among users versus non-users in the elderly (≥65) (36.4% vs. 42.4%, *p* = 0.016) and non-Hispanic white subgroups (28.9% vs. 38.3%, *p* < 0.001). There were no significant PC prevalence differences between users and non-users in the younger population (50–64) or other race/ethnicity. In the multivariable analyses, former marijuana use was associated with lower PC compared to never using (odd ratio = 0.74, 95% CI 0.62–0.90, *p* = 0.001). Current use was also suggestive of reduced prevalence but was not statistically significant (odd ratio = 0.77, 95% CI 0.52–1.14, *p* = 0.198), possibly due to low sample size. Our findings from a large national survey provide additional data to link marijuana use with lower PC prevalence.

## 1. Introduction

Prostate cancer (PC) is the second most common cancer and the fifth leading cause of cancer-associated death in males worldwide, with an estimated 1.4 million new cases and 375,000 deaths worldwide per year [1]. Factors such as age, African-American ethnicity, and certain genetic factors are known risk factors for PC [2]. However, little is known about factors that may have protective effect against PC.

An understudied area is the putative reduced risk for PC related to regular use of cannabis (i.e., marijuana). Several preclinical data from cell line and animal xenograft models have demonstrated anti-tumor effects of cannabinoids in PC [3,4]. WIN 55,212 (a cannabinoid receptor agonist) reduced proliferation and arrested cells in the G0/G1 phase via CB2 receptor-dependent signaling in prostate cancer cells [5,6]. Additionally, PM49 (a synthetic cannabinoid quinone) was found to result in significant inhibition of tumor growth in xenograft PC models [3,7].

Numerous mechanisms of the anticancer effects of cannabinoids have been proposed, including (1) dysregulation of the cell cycle via inhibition of cyclin–CDK complexes, cAMP, AKT pathways, and downregulation of Cdc2; (2) proapoptotic effects through enhanced ROS generation, activation of caspase8-9, inhibition of PI3K/Akt and RAS-MAPK/ERK pathways; and (3) proautophagic effects via accumulation of ceramide, inactivation of mTORC1, and activation of LC3-II through cannabinoid receptors CB1 or CB2 activation, or through CB1,2 independent manners [5,8]. In addition, other mechanisms such as (4) anti-invasive, (5) anti-angiogenic, and (6) anti-metastatic effects via different signaling pathways have been suggested as anticancer effects of cannabinoids [5,8]. Additionally, a few retrospective human studies have suggested that marijuana use may be associated with reduced risk of other solid tumors such as bladder, head, and neck cancer [9,10], although a few others have suggested the opposite. These observations prompted us to undertake a study to investigate the potential antineoplastic role of cannabinoids in patients with PC.

To our knowledge, there have been no human studies examining the relationship between marijuana use and PC. We performed a cross-sectional study to investigate the association of marijuana use and the risk of PC using data from the National Survey on Drug Use and Health (NSDUH) survey administered by the U.S. Department of Health and Human Services.

## 2. Materials and Methods

### 2.1. Data Source

Our study utilized the public data from 2002–2020 from NSDUH, a cross-sectional US-representative survey administered by the Department of Health and Human Services consisting of noninstitutionalized persons aged twelve and older in 50 states and District of Columbia since 1971 [11]. In each NSDUH survey, household addresses are randomly selected for participation through multistage sampling. NSDUH conducts interviews with selected individuals within a household. Individuals are surveyed at a single time point, in their home online via a computer and/or in-person with assistance from a NSDUH interviewer. Participants receive $30 (USD) compensation for completion of the interview.

Inclusion criteria for the current analysis were being male, aged fifty years or older, and self-reports of ever having testicular/prostate cancer at the time of interview. We excluded patients below the age of fifty since the incidence rate of prostate cancer in this age group is very low (1 in 350–450) [12,13]. This age cut-off also allowed us to exclude the vast majority of patients with testicular cancer, an uncommon cancer with very low incidence (5.9 per 100,000 men) and predominantly seen in young adults less than forty-five years of age [14].

### 2.2. Measures

#### 2.2.1. Dependent Variable (DV): Prostate Cancer (Yes, No)

Respondents were asked whether they had ever been told by a doctor or other medical professional that they have cancer and if yes, then they were asked “was your cancer type testicular/prostate cancer.” Their responses were recoded into binary categories as having or not having a testicular/prostate cancer diagnosis. Although the question sought information on diagnosis of either testicular or prostate cancer from participants, given the low prevalence rate of PC in men under the age of forty-five [12] and overall rarity of testicular cancer in the general population especially among those above the age of fifty [14], misclassification of participants with PC from those with testicular cancer may be negligible in our study sample.

#### 2.2.2. Independent Variable (IV): Marijuana Use

Marijuana use: Marijuana use was measured by asking respondents, “Ever used marijuana/hashish” and “Time since last used marijuana/hashish”. Participants’ responses were classified into three categories: (1) current use for those who consumed any marijuana during the last thirty days at the time of interview; (2) former use for those who consumed any marijuana between more than thirty days from the time of interview and sometime in their lifetime; (3) never use for those who reported no use of any marijuana products in their lifetime. Other marijuana-related variables evaluated in the study included age at first use, number of days used in the past thirty days, use of medical marijuana, and legality of marijuana use in state of residence.

#### 2.2.3. Confounders and Covariates

We examined the following potential confounders (i.e., related to both cancer risk and marijuana use): tobacco use and alcohol consumption which were categorized in a similar fashion as marijuana use (i.e., current, former, never). For tobacco use, an additional variable among cigarette smokers was measured by asking if the respondents smoked more than a hundred cigarettes in their lifetime (i.e., ever versus never). Race/ethnicity contains non-Hispanic white (NHW), non-Hispanic black (NHB), Hispanic, and non-Hispanic others including native American/Alaskan native, non-Hispanic native Hawaiian/other pacific islanders, non-Hispanic Asian, and non-Hispanic more than one race. The other background covariates include age (50–64 years and >65 years), education (less than high school, high school, associate degree, and college graduate), marital status (single, married, widowed, and divorced), ever served in armed forces service, urbanicity, annual income, and insurance status (Medicare, private insurance, state health insurance, and special insurance through military including CHAMPUS or TRICARE, CHAMPVA, the VA, or other federally sponsored health care).

### 2.3. Statistical Analyses

We performed Chi-Square tests comparing the frequencies of the independent variables and potential confounders/covariates according to the PC group (yes, no). For continuous variables (e.g., age of the first use of marijuana, total number of days marijuana used in the last thirty days or in the last twelve months), we conducted *t*-tests for independent samples using the group factor of PC status.

We performed a stratified Cochrane–Mantal–Haenszel test (CMH) to test for the association of having PC with marijuana use while taking into account age and race/ethnicity groups. When the associations were determined to be similar across the groups according to the Breslow–Day (BD) test, we estimated the common odds ratio (OR) along with 95th percentile confidence intervals.

These findings as well as the descriptive analyses, described above, were applied to construct an elastic net regularization for variable selection [15]. After the variables were selected, we conducted a multivariable logistic model (see definition of IVs and DVs above) for selected variables as predictors and calculated their odds ratio, confidence interval, and *p*-values. All analyses including multivariable analysis were conducted using the base “R” software v4.1.0 (R Core Team, 2021) and “glmnet” package with a *p*-value of <0.05 treated as statistically significant.

## 3. Results

### 3.1. Sample Description

In this sample of 2503 males aged greater than fifty years (Table 1), most participants were age sixty-five years and older (69%), married (60.9%), had some college education (73%), reported an annual income over $50,000 (64%), and lived in a metropolitan area (77.8%), and nearly all reported having some type of medical insurance. Forty percent reported having served in the armed forces. Lastly, 89.8% of participants were non-Hispanic white (NHW), 4.8% non-Hispanic black (NHB), 3.2% Hispanic, and 2.3% non-Hispanic others (NH other).

### 3.2. Prostate Cancer Prevalence

Thirty-six percent (903/2503) reported having a diagnosis of PC at the time of interview (Table 1). Compared to the participants without PC diagnosis, those with PC were more likely to be ≥65 years (64.9%, 77.2%, respectively, *p* < 0.001) and less likely to be non-Hispanic white (92.8%, 84.4%, respectively, *p* < 0.001). Looking at racial/ethnic differences, 68% non-Hispanic blacks reported having a diagnosis of PC, followed by Hispanics (48%), and then non-Hispanic whites (33.9%). (*p* < 0.001). Further, a higher rate (40.2%) of having a PC diagnosis was found in those aged sixty-five years and over, compared to 26.8% in those under sixty-five (Figure 1). Specifically, among those sixty-five years and older, 77% (95% CI: 65.6, 85.1) among non-Hispanic blacks, 52% (95% CI: 41.3, 67.7) among Hispanics, and 38% (95% CI: 35.7, 40.5) non-Hispanic whites reported having a PC diagnosis. As stated above, black participants in the study were only 4.8% (=120/2503) of the total study sample, but they constituted 9.1% (=82/903) of all patients who reported having PC diagnosis. In contrast, non-Hispanic whites made up 89.8% (=2247/2503) of the total study participants but constituted 84.4% (=762/903) among those reported to have a diagnosis of PC.

### 3.3. Prevalence of Marijuana and Other Substance Use

In the full sample, 53.4% reported never having used marijuana, 40.8% identified as former users, and 5.8% reported current use (Table 2). When examining marijuana use by PC status, we observed more non-users among those with the malignancy compared to controls (59.1%, 50.2%, respectively, *p* < 0.001). A significantly lower prevalence of PC was noted among current marijuana users (46/145, 31.7%) and former marijuana users (323/1021, 31.6%) in comparison to the non-users (534/1337, 39.9%, *p* < 0.001). Participants with a diagnosis of PC reported starting marijuana use at a slightly older age compared to the non-prostate controls (22.2 yrs. vs. 21.1 yrs.; *p* = 0.04). No other marijuana-related risk factors in Table 2 (e.g., total number of days used in past 30 days) or other substance use (tobacco and alcohol) showed significant associations with PC status.

Table 3 shows lower rates of PC diagnosis among marijuana users compared to the non-users in both age groups with no significant difference in odds ratios (BD test *p* = 0.719). The combined effect across age groups showed 21% reduction in having PC among marijuana users compared to that among non-users (OR_CMH_ = 0.79, 95% CI: 0.67, 0.94, *p* = 0.008). More specifically, there was a 22% reduction in having PC among older marijuana users aged 65 years (OR = 0.78, 95% CI: 0.64, 0.95, *p* = 0.016) compared to their counterpart nonusers. Such an effect was not significantly observed in the younger age group of 50–64 (Table 3, Figure 1).

Considering the racial/ethnic differences in rates of having PC, there was no significant difference in odds ratios (BD test *p* = 0.93). The combined effect across racial/ethnic groups showed a 32% reduction in having PC among marijuana users compared to that among non-users (OR_CMH_ = 0.68, 95% CI: 0.57, 0.81, *p* < 0.001). However, within each racial group, only white marijuana users showed a 35% significant reduction in having PC when compared to the white non-users (OR = 0.65, 95% CI: 0.55, 0.78, *p* < 0.001). Such a significant effect was not observed in other racial/ethnic groups, possibly due to small subgroup samples. Thus, in general, regardless of age or racial/ethnic status, marijuana users had a significant reduction in having prostate PCthan non-users.

Multivariable logistic regression analyses (Table 4) suggested that former marijuana users in the survey reported a significantly lower prevalence of PC compared to non-users (OR = 0.74, 95% CI 0.62–0.90, *p* = 0.001). Additionally, self-reported rates of PC among current users also trended towards a lower prevalence but did not reach statistical significance (OR = 0.77, 95% CI 0.52–1.14, *p* = 0.198) possibly due to low sample size. As expected, older age (OR = 2.31, 95% CI 1.65–3.26) and non-Hispanic black ethnicity (OR = 5.63, 95% CI 1.65–3.26) were associated with higher prevalence of PC in relation to their respective referent groups (<65 years; and non-Hispanic whites). In addition, participants who had Medicare/private insurance, private insurance only, or special insurance only were more likely to report having PC when compared to those with other insurance.

## 4. Discussion

In this cross-sectional study of 2503 participants from the USA using the NSDUH survey (2002 to 2020), we observed that individuals who were former marijuana users had a significantly lower rate of self-reports of having PC. Additionally, the current marijuana users also trended towards lower self-reports of PC. The lack of statistical significance in the analysis of current marijuana users is likely due to a low sample size of the groups with and without PC (i.e., 46 and 99, respectively). Consistent with well-established risk factors, our analyses also found that older participants (≥65 years) and non-Hispanic black participants had a higher prevalence of PC, providing credibility to our overall study. In subgroup analyses, the patterns observed in the full sample were maintained. Specifically, among participants aged greater than 65 years, former marijuana use was linked to reduced self-reports of PC compared to never using. Similarly, among non-Hispanic whites exclusively, former marijuana use was associated with lower rates of self-reported PC compared to never use.

Increased rates of PC in black individuals can be attributed to an interplay of genetic, healthcare, and socioeconomic status-related factors. Genetic susceptibility through variations in the androgen receptor gene have been associated with an increased risk of PC in black males of African-American ancestry [16]. Additionally, distinct genomic alterations reported including metabolic dysregulation and inflammatory and cytokine signaling relative to white men have been identified in prostate tumors of black individuals that may contribute to more aggressive disease [17,18]. Healthcare-related factors driven by a culture of mistrust in the healthcare system, a lack of strong physician–patient relationships due to poor communication, and a lack of information on PC and limited access to treatment options result in the disparities in outcomes [19]. This is further exacerbated by other social determinants of health including socioeconomic status (SES) and environmental exposures leading to differences in incidence and outcomes of PC through both biological mechanisms (inflammation and cell damage) and healthcare system inequities [20].

Our findings provide corroborative data from a large national, population-based survey to strengthen the existing body of evidence suggesting a potentially protective role of marijuana against the development of PC. To our knowledge, very few studies have investigated the association between marijuana use and PC. A large retrospective cohort study from California revealed marijuana use to be associated with increased risk of PC (relative risk = 3.1, 95% CI = 1.0–9.5) among tobacco nonsmokers [21]. However, the study population was aged between fifteen to forty-nine years. Further, the risk of PC in marijuana users was not significant after adjustment for cigarette smoking status [21]. Further, a metanalysis analyzing the association between marijuana and cancer reported that this study had a moderate risk of bias [22]. One of the strengths of our study is that it focuses on individuals greater than fifty years of age, who have a significantly higher incidence of PC.

Another strength of our findings is biologic support for the anti-cancer effects of the constituents of marijuana. Cannabinoids have been proposed as a regulator for cancer cell growth, differentiation, invasion, and metastasis in multiple mechanisms [3,23]. Cannabinoid receptors regulate the mitogen-activated protein kinases (MAPKs) pathway, which involves cell proliferation, differentiation, and apoptosis [3,24]. Cannabinoids also stimulate the p8-regulated pathway, which induces autophagy and blocks the activation of the VEGF pathway [23]. Several pre-clinical studies have demonstrated the anti-tumor effect of cannabinoids in PC. Roberto et al. showed WIN 55,212-2, a synthetic cannabinoid, inhibits PC cell growth and proliferation, spread, and invasion. Furthermore, in the PCcell line models, it has also been documented to induce cell cycle arrest and promote apoptosis [6]. Recently, a systemic review with six vivo-xenograft mouse model studies reported both synthetic and natural cannabinoids have anti-cancer effects, including a reduction in cell proliferation, tumor size, and survival benefit [3,25]. Several observational studies suggested that marijuana use may be associated with a reduced risk of other solid cancers including bladder, head, and neck cancer [9,10]. Recent meta-analyses have shown heterogeneous results with regards to the association between marijuana use and cancer risk across different tumor types. One meta-analysis, comprising twenty-five observational studies, indicated low-strength evidence suggesting an association between marijuana use and the increased development of testicular germ cell tumors [22]. However, the association of marijuana with other cancer types remained unclear and was limited by low exposure and short duration of follow-up [22]. Another meta-analysis, encompassing thirty-four observational studies revealed a trend with negative cancer association between marijuana use and non-testicular malignancies [26]. Cannabis use showed a significant negative association with head and neck cancers (RR = 0.83, *p* < 0.05) and non-testicular cancers (RR = 0.87, 95% CI = 0.78–0.98, *n*= 41, *p* < 0.025) [26]. Notably, both meta-analyses only included one retrospective cohort study involving PC, and as mentioned earlier, had limited representation of elderly groups and exhibited a moderate risk of bias [21,22,26]. This further highlights the importance of our study with focus on the at-risk group while investigating the relationship between marijuana use and the risk of PC. Since medical marijuana is being used more frequently in cancer patients for pain control, nausea, and abdominal pain [27] and nearly half of oncologists report prescribing medical marijuana to patients at some point in their practice [28], future prospective studies in patients on medical marijuana may facilitate our further understanding of potential anticancer properties.

Our study is not without limitations. The study population derived from the NSDUH survey is US-based and is largely made up of non-Hispanic white participants with few participants from other ethnicities. This can limit the impact of the generalizability of the results of the study. We did, however, find an increased prevalence of PC among non-Hispanic blacks, which is consistent with long-term population-based evidence. The PC prevalence of 36% in this study is significantly higher than that of the general male elderly population, which can potentially imply selection bias of the study group [29]. Another limitation is that the NSDUH survey question enquiring about cancer history includes both testicular or PC as a single choice. Our choice of limiting the study sample to those aged greater than fifty or over mitigated this problem due to the very low rates of testicular cancer in this age group [14]. Another limitation is that the history of other cancers is unknown in the control group. Additionally, dose-dependent effects of marijuana could not be ascertained because neither a standard of amount per use nor type of marijuana were captured in the NSDUH database. For example, recreational marijuana, unlike medical or synthetic marijuana, is known to contain over hundred phytocannabinoids and thousands of other components [30]. Thus, there would be significant heterogeneity between strains of recreational marijuana components as well as between their putative biologic effects. Lastly, a major constraint of our analyses is the nature of cross-sectional study in which temporal causality cannot be established.

## 5. Conclusions

To our knowledge, this is the first study investigating the association between marijuana use and PC in a large cohort, using a national survey focused on the at-risk group of the older male population. Our findings can serve as hypothesis-generating for future prospective studies to further evaluate the role of cannabinoids (using medical marijuana) in PC prevention.

## Figures and Tables

**Figure 1 biomedicines-12-01008-f001:**
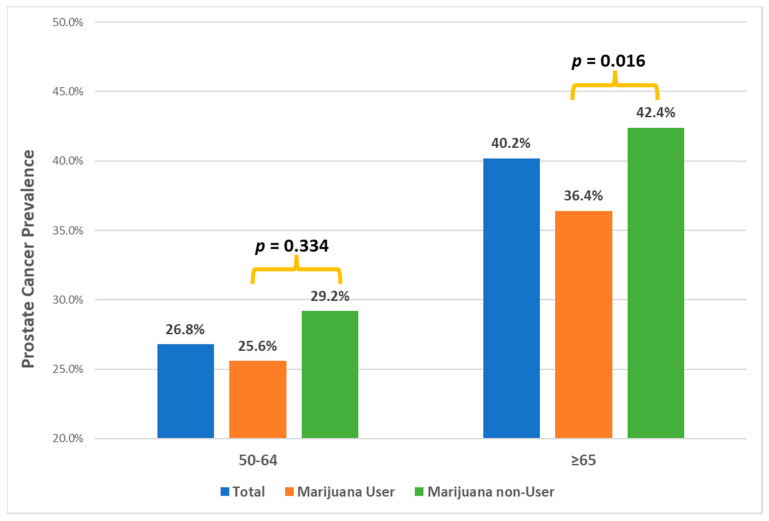
Prostate cancer prevalence by marijuana usage and age.

**Table 1 biomedicines-12-01008-t001:** Demographic characteristics in relation to prostate cancer, NSDUH surveys: 2002–2020 (*n* = 2503).

		Prostate Cancer		
Characteristics	Total (*n* = 2503)	Yes (*n* = 903)	No (*n* = 1600)	*p*-Value
	*n* (%)	*n* (%)	*n* (%)	
Age (in years)				
50–64	768 (30.7%)	206 (22.8%)	562 (35.1%)	<0.001
≥65	1735 (69.3%)	697 (77.2%)	1038 (64.9%)	
Race/Ethnicity				
Non-Hispanic white	2247 (89.8%)	762 (84.4%)	1485 (92.8%)	<0.001
Non-Hispanic black	120 (4.8%)	82 (9.1%)	38 (2.4%)	
Hispanic	79 (3.2%)	38 (4.2%)	41 (2.6%)	
Non-Hispanic others	57 (2.3%)	21 (2.3%)	36 (2.3%)	
Education				
<High school degree	178 (7.1%)	64 (7.1%)	114 (7.1%)	0.99
=High school degree	532 (21.3%)	196 (21.7%)	336 (21.0%)	
Some college/associate degree	635 (25.4%)	230 (25.5%)	405 (25.3%)	
College graduate and above	1158 (46.3%)	413 (45.7%)	745 (46.6%)	
Marital status				
Married	1518 (60.6%)	541 (59.9%)	977 (61.1%)	0.10
Widowed	201 (8.0%)	89 (9.9%)	112 (7.0%)	
Divorced/Separated	251 (10.0%)	77 (8.5%)	174 (10.9%)	
Never married	129 (5.2%)	38 (4.2%)	91 (5.7%)	
Missing	404 (16.1%)	158 (17.5%)	246 (15.4%)	
Total family income (in $)				
<20,000	197 (7.9%)	62 (6.9%)	135 (8.4%)	0.63
20,000–49,999	706 (28.2%)	259 (28.7%)	447 (27.9%)	
50,000–74,999	453 (18.1%)	178 (19.7%)	275 (17.2%)	
75,000 or more	1147 (45.8%)	404 (44.7%)	743 (46.4%)	
Insurance status				
Medicare and private insurance	1155 (46.1%)	460 (50.9%)	695 (43.4%)	0.001
Medicare only	580 (23.2%)	214 (23.7%)	366 (22.9%)	
Private insurance only	640 (25.6%)	197 (21.8%)	443 (27.7%)	
Special insurance only *	45 (1.8%)	18 (2.0%)	27 (1.7%)	
Others **	83 (3.3%)	14 (1.6%)	69 (4.3%)	
Residence status				
Metro	1947 (77.8%)	710 (78.6%)	1237 (77.3%)	0.74
Rural	556 (22.2%)	193 (21.4%)	363 (22.7%)	
History of Diabetes				
**Yes**	542 (21.7%)	195 (21.6%)	347 (21.7%)	0.99
**No**	1961 (78.3%)	708 (78.4%)	1253 (78.3%)	
Served in armed forces				
Yes	742 (41.6%)	400 (44.3%)	642 (40.1%)	0.25
No	1444 (57.7%)	495 (54.8%)	949 (59.3%)	
Missing	17 (0.7%)	8 (0.9%)	9 (0.6%)	

* Special insurance includes CHAMPUS or TRICARE, CHAMPVA, the VA, or other military health care. ** Others include state health insurance.

**Table 2 biomedicines-12-01008-t002:** Description of usage patterns of substances including cigarette, alcohol, and marijuana among study participants.

		Having Prostate Cancer		
Substance Use	Total (*n* = 2503)	Yes (*n* = 903)	No (*n* = 1600)	*p*-Value
	*n* (%)	*n* (%)	*n* (%)	
Marijuana use *				<0.001
Current	145 (5.8%)	46 (5.1%)	99 (6.2%)	
Former	1021 (40.8%)	323 (35.8%)	698 (43.6%)	
Never	1337 (53.4%)	534 (59.1%)	803 (50.2%)	
Cigarette smoking ^$^				0.11
Current	1227 (49.0%)	434 (48.1%)	793 (49.6%)	
Former	596 (23.8%)	197 (21.8%)	399 (24.9%)	
Never	680 (27.2%)	272 (29.9%)	408 (25.5%)	
Alcohol consumption ^ɣ^				0.82
Current	1654 (66.1%)	606 (67.1%)	1048 (65.5%)	
Former	689 (27.5%)	236 (26.1%)	453 (28.3%)	
Never	160 (6.4%)	61 (6.8%)	99 (6.2%)	
Age (Mean, SD) of first marijuana use (*n* = 1159)	21.5 (8.5)	22.2 (8.6)	21.1 (8.4)	0.04
Total number of days (mean, SD) marijuana used in last 30 days (*n* = 145)	14.2 (11.2)	13.5 (11.2)	14.6 (11.2)	0.93
Total number of days (mean, SD) marijuana used in last 12-months (*n* = 208)	122 (127)	121 (129)	123(126)	0.60
Used medical marijuana in past 12 months (*n* = 216)				
Yes	30 (1.4%)	8 (1.1%)	22 (1.6%)	0.75
No	141 (6.7%)	46 (6.1%)	95 (7.1%)	
State medical marijuana law in place at time of interview				
Yes	1556 (62.6%)	568 (62.9%)	998 (62.4%)	0.60
No	937 (37.4%)	335 (37.1%)	602 (37.6%)	

* “Current” marijuana user includes individuals who used marijuana in the last 30 days; “Former” marijuana user includes those who smoked marijuana some time in their life excluding the past 30 days; “Never” smoker is an individual who never smoked marijuana in their lifetime. ^$^ “Current” smoker includes individuals who smoked cigarettes in the last 30 days; “Former” smoker includes those who smoked cigarettes some time in their life excluding the past 30 days; “Never” smoker is an individual who never smoked a cigarette in their lifetime. ^ɣ^ “Current” alcohol drinker includes individuals who consumed alcohol within the last year; “Former” alcohol drinker includes those who drank alcohol some time in their life excluding the past year; “Never” drinker includes those who never consumed alcohol in their lifetime.

**Table 3 biomedicines-12-01008-t003:** Distribution of prostate cancer status by marijuana use and age or race (*n* = 2503).

Marijuana Use		Total	Had Prostate Cancer	OR	95% CI	*p*-Value
			*n* (%)			
Age (in years)						
50–64	Yes	518	133 (25.6%)	0.84	(0.60, 1.17)	0.344
	No	250	73 (29.2%)			
≥65	Yes	648	236 (36.4%)	0.78	(0.64, 0.95)	0.016
	No	1087	461 (42.4%)			
Race/Ethnicity						
NH black	Yes	63	41 (65.1%)	0.73	(0.31, 1.69)	0.542
	No	57	41 (71.9%)			
Hispanic	Yes	29	15 (51.7%)	1.25	(0.46, 3.38)	0.797
	No	50	23 (46.0%)			
NH white	Yes	1043	301 (28.9%)	0.65	(0.55, 0.78)	<0.001
	No	1204	461(38.3%)			
NH other	Yes	31	12 (38.7%)	1.19	(0.35, 4.08)	0.965
	No	26	9 (34.6%)			

From the Cochran–Mantal–Haenszel test, the overall effect of marijuana use is about 21% reduction of prostate cancer controlling for age: OR = 0.79, 95% CI: 0.67, 0.94; *p* = 0.008. From the Cochran–Mantal–Haenszel test, the overall effect of marijuana use is about a 32% reduction in prostate cancer controlling for race: OR = 0.68, 95% CI: 0.57, 0.81; *p* < 0.001.

**Table 4 biomedicines-12-01008-t004:** Multivariate logistic regression of odds for having prostate cancer in relation to marijuana use, alcohol consumption, and demographic factors.

Variable	Odds Ratio	95% CI	*p*-Value
Marijuana use			
Current	0.77	0.52–1.14	0.198
Former	0.74	0.62–0.90	0.001 *
Never user	Referent	--	-
Age (years)			
>65	2.31	1.65–3.26	<0.001 *
50–64	Referent	--	
Race/Ethnicity			
Non-Hispanic Black	5.63	3.75–8.62	<0.001 *
Hispanic	2.11	1.32–3.37	0.002 *
Non-Hispanic Other	1.34	0.75–2.34	0.308
Non-Hispanic White	Referent	--	
Alcohol consumption			
Current	1.36	0.95–1.96	0.096
Former	1.03	0.71–1.51	0.871
Never	Referent	--	-
Insurance status			
Medicare and private	2.01	1.03–4.11	0.046 *
Medicare only	1.82	0.93–3.72	0.089
Private insurance only	2.74	1.48–5.42	0.002 *
Special insurance only	3.07	1.27–7.57	0.013 *
Other	Referent	--	

* Conveys statistical significance (*p* < 0.05).

## Data Availability

These data were derived from the following resources available in the public domain: https://www.samhsa.gov/data/report/2020-nsduh-detailed-tables. “URL accessed on 2 December 2023”.

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
