# Peer review of "Marijuana Use May Be Associated with Reduced Prevalence of Prostate Cancer: A National Survey on Drug Use and Health Study from United States of America"

_biomedicines, 2024, doi:10.3390/biomedicines12051008_

Round 1

Reviewer 1 Report

Comments and Suggestions for Authors

Describing "reports that cannabis use may be associated with lower prostate cancer prevalence," the paper showed that among participants over 65, self-reported prostate cancer was reduced among those who had ever used cannabis compared with those who had never used it. Similarly, among non-Hispanic whites alone, those who had ever used marijuana had lower rates of self-reported prostate cancer compared to those who had never used marijuana. This study can provide hypotheses for future prospective studies of prostate cancer treatment and has certain important significance. However, the following questions should be noted before publication:

1. It is suggested to add supplementary materials to show the correlation analysis in the questionnaire.

2. Do the patients who use marijuana have any other medical conditions, and do these other medical conditions lead to a lower risk of prostate cancer?

3. Why are black men more likely to have prostate cancer than other men? The text doesn't seem to explain.

4. Is there a definite relationship between participants with special insurance and prostate cancer?

5. In Table 2, the number of people who use cigarettes and alcohol and do not get sick is higher than the number of people who get sick. Can it be shown that cannabis is related to the reduction of prostate cancer prevalence?

6. In the results discussed, it does not seem reasonable to suggest that only a small group of participants who had ever used cannabis had a lower risk of prostate cancer than those who had never used cannabis.

7. Among cigarette smokers, the risk of prostate cancer is not significant among marijuana users. Are there some components in cigarettes that can cause this?

Author Response

Dear Reviewer,

Reviewer 2 Report

Comments and Suggestions for Authors

In the submitted paper the association of marijuana use and prostate cancer is explored, based on data obtained from a national survey. The country considered for this study was the United States of America. Although the analysis was performed with data from a specific country and corresponding to a specific period of time, the results obtained can be relevant for other countries.

Bellow I present some specific comments:

Lines 1 to 4: The country should be specified.

Lines 26 to 27: The country should be specified.

Line 34: Replace “multivariate” by “multivariable”, as there is just one dependent variable.

Line 37: I suggest the replacement of the word “low” with “lower”.

Lines 127 to 129: Replace “multivariate” by “multivariable”. The authors state that the findings of the statistical analyses presented in the previous paragraphs were “applied to construct a more parsimonious multivariate logistic model”. A more detailed presentation of the multivariable analysis performed should be given, including the steps considered to construct this logistic model, and what is considered by the authors to be a “parsimonious” model.

Lines 211, 222: Replace “multivariate” by “multivariable”.

Lines 225 to 228: The sentence is not clear. I suggest the removal of “and possibly current” from this first sentence, and the inclusion of the tendency found in the analysis of the current marijuana use in the second sentence. Include also the country where the study took place.

Lines 229 to 231: Verify the writing.

Lines 266 to 283: In this paragraph where the limitations are presented, the implications of the use of a national survey from a specific country should be included.

Author Response

Dear Reviewer,

Round 2

Reviewer 1 Report

Comments and Suggestions for Authors The manuscript, which describes "reports that cannabis use may be associated with reduced prostate cancer prevalence," investigates the relationship between cannabis use and prostate cancer, focusing on high-risk groups in the older male population. The findings of the study can provide hypotheses for future prospective studies to further evaluate the role of cannabinoids (use of medical marijuana) in prostate cancer prevention, which has certain significance. There were no obvious errors in the article, which provided enough background information and sufficient data to support the rationality of the investigation report, showing the prospective use of marijuana in the medical field and providing innovative ideas for future medical treatment.

Reviewer 2 Report

Comments and Suggestions for Authors

The authors have made all the adjustments and corrections requested.